Subject Areas:
evolution, ecology

Keywords:
adaptive decoupling, metamorphosis, complex life cycles, ontogenetic niche shifts, disparity, geometric morphometrics

Author for correspondence:
Peter T. Rühr
e-mail: ruehr@uni-bonn

# Juvenile ecology drives adult morphology in two insect orders

Peter T. Rühr[1], Thomas van de Kamp[2,3], Tomáš Faragó[3], Jörg U. Hammel[4], Fabian Wilde[4], Elena Borisova[5], Carina Edel[1], Melina Frenzel[1], Tilo Baumbach[2,3] and Alexander Blanke[1,6]

[1]Institute of Evolutionary Biology and Animal Ecology, University of Bonn, An der Immenburg 1, 53121 Bonn, Germany
[2]Institute for Photon Science and Synchrotron Radiation (IPS), Karlsruhe Institute of Technology (KIT), Hermann-von-Helmholtz-Platz 1, 76344 Eggenstein-Leopoldshafen, Germany
[3]Laboratory for Applications of Synchrotron Radiation (LAS), Karlsruhe Institute of Technology (KIT), Kaiserstr. 12, 76131 Karlsruhe, Germany
[4]Institute of Materials Physics, Helmholtz-Zentrum Hereon, Max-Planck-Straße 1, 21502 Geesthacht, Germany
[5]Swiss Light Source, Paul Scherrer Institute, Forschungsstrasse 111, 5232 Villigen, Switzerland
[6]Medical and Biological Engineering Research Group, School of Engineering and Computer Science, University of Hull, Hull HU6 7RX, UK

  PTR, 0000-0003-2776-6172; TvdK, 0000-0001-7390-1318; JUH, 0000-0002-6744-6811; CE, 0000-0003-1760-6669; MF, 0000-0002-5264-6319; AB, 0000-0003-4385-6039

Most animals undergo ecological niche shifts between distinct life phases, but such shifts can result in adaptive conflicts of phenotypic traits. Metamorphosis can reduce these conflicts by breaking up trait correlations, allowing each life phase to independently adapt to its ecological niche. This process is called adaptive decoupling. It is, however, yet unknown to what extent adaptive decoupling is realized on a macroevolutionary scale in hemimetabolous insects and if the degree of adaptive decoupling is correlated with the strength of ontogenetic niche shifts. It is also unclear whether the degree of adaptive decoupling is correlated with phenotypic disparity. Here, we quantify nymphal and adult trait correlations in 219 species across the whole phylogeny of earwigs and stoneflies to test whether juvenile and adult traits are decoupled from each other. We demonstrate that adult head morphology is largely driven by nymphal ecology, and that adult head shape disparity has increased with stronger ontogenetic niche shifts in some stonefly lineages. Our findings implicate that the hemimetabolan metamorphosis in earwigs and stoneflies does not allow for high degrees of adaptive decoupling, and that high phenotypic disparity can even be realized when the evolution of distinct life phases is coupled.

## 1. Background

Species ecology can change dramatically during development [1,2], a process called 'ontogenetic niche shift' [3]. If phenotypic traits are coupled between life phases, ontogenetic niche shifts may result in adaptive conflicts, because coupled traits cannot evolve independently according to divergent, life-phase-specific needs [2,4,5]. Metamorphosis, a process of rapid change in morphology, physiology and behaviour [6,7], can break up trait correlations between life phases [5,8–12] and therefore allows for an adaptive decoupling of traits [4,13,14]. Even though around 80% of animals, including all winged insect lineages (Pterygota), show complex life cycles with metamorphic periods [13], the macroevolutionary relationship between ontogenetic niche shifts and adaptive decoupling has only been studied in Echinodermata [9,10,15], ray-finned fishes [16], frogs [17–19] and salamanders [20]. In these groups, it has been shown

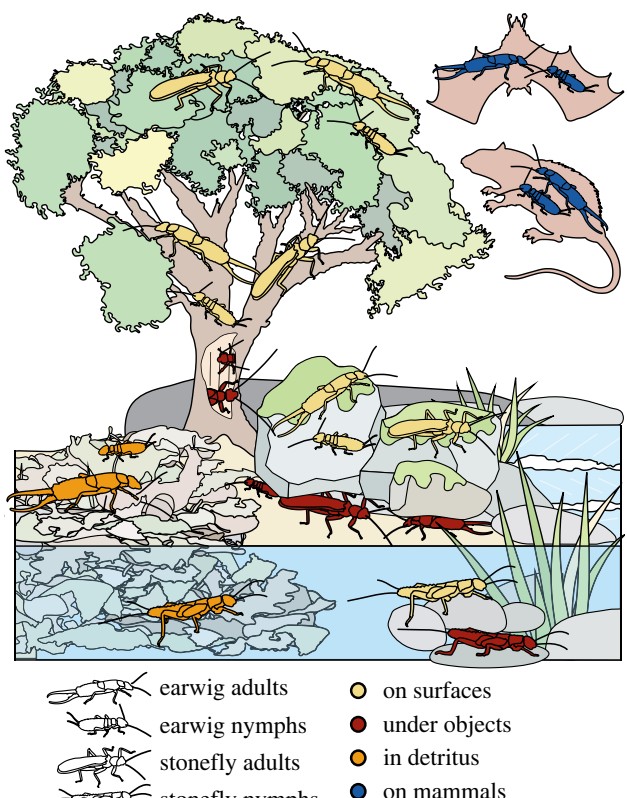

**Figure 1.** Overview of microhabitat occupation by nymphal and adult earwigs and stoneflies. Insect pictograms indicate different taxa and life stages. Habitat preference colour-coded. (Online version in colour.)

Legend:
- earwig adults
- earwig nymphs
- stonefly adults
- stonefly nymphs
- on surfaces
- under objects
- in detritus
- on mammals

that adaptive decoupling via metamorphosis seems to largely allow independent phenotypic evolution in distinct life-history phases. In insects, however, no macroevolutionary assessment of adaptive decoupling has been undertaken, so it remains unknown to what extent the nymphal life phase influences the phenotypic evolution of the adult phase.

Winged insects show two types of metamorphosis: hemimetaboly and holometaboly. Holometabolous insects possess a pupal stage between the adult and the juvenile life phase in which strong morphological remodelling takes place ('complete metamorphosis' [7,11]). While some traits may still be linked [21–24], general morphology, feeding behaviour, and modes of locomotion of holometabolous juveniles and their adults are usually strongly divergent [4,7]. The separation of life phases by complete metamorphosis has been hypothesized as a key innovation driving insect diversification [25,26]. Hemimetabolous insects, which gradually develop towards adulthood without a pupal stage, can also show strong ontogenetic niche shifts between the juvenile and the adult phases. For example, dragonflies (Odonata), mayflies (Ephemeroptera) and stoneflies (Plecoptera) shift from aquatic to terrestrial habitats, and some stonefly species develop from predatory nymphs to non-feeding adults. Yet, considerable morphological changes during this step mostly affect wings, genitals and gills, while the general body plans of nymphs and their adults remain relatively similar (hemimetabolan metamorphosis [7]).

Here, we used three-dimensional geometric morphometrics, a de novo generated database on ecological traits and multivariate statistics to investigate macroevolutionary correlations between nymphal ecology and adult shape. We hypothesize that (i) the single moult between the juvenile and the adult life phase might not be able to allow for high degrees of adaptive decoupling, (ii) adaptive decoupling is

stronger in taxa with stronger ontogenetic niche shifts, and (iii) the independent evolution of life phases in taxa with strong adaptive decoupling results in diversification into more ecological niches and thus a higher shape disparity.

A statistically rigorous approach required selecting two morphologically similar lineages with different ecologies, which are as closely related as possible to allow a meaningful correction of potentially biasing phylogenetic signal [27–29]. We chose to study the two closely related [30,31] hemimetabolous insect orders earwigs (Polyneoptera: Dermaptera, approx. 2000 described species [32]) and stoneflies (Polyneoptera: Plecoptera, approx. 3400 species [32]). Earwigs and stoneflies both possess prognathous biting–chewing mouthparts, similar antennal and eye positions and sizes, a low performance flight apparatus, and exhibit a similar size range with body lengths mostly between 10 and 30 mm [33,34]. However, earwigs and stoneflies differ in the degree of ontogenetic niche shift: earwig nymphs and adults are both terrestrial (figure 1) and feed on similar food sources [33], while stoneflies, with few secondarily evolved exceptions (e.g. [35,36]), are the only polyneopterans that show an amphibiotic lifestyle with aquatic nymphs and terrestrial adults (figure 1) [34]. They transition from an aquatic to a terrestrial environment over the course of the final moult, often accompanied by a shift in feeding mode [37].

## 2. Material and methods

### (a) Taxon sampling

We studied 219 species (electronic supplementary material, table S10), 144 earwigs and 75 stoneflies, covering all extant families, 80.3% of extant subfamilies and 32.28% of extant genera [38,39]. Specimens were loaned from the Natural History Museum (NHMUK) (London, UK), Museum für Naturkunde (MfN) (Berlin, Germany), Zoologisches Forschungsmuseum Alexander Koenig (ZFMK) (Bonn, Germany), Muséum national d'Histoire naturelle (MNHM) (Paris, France), Naturhistorisches Museum (NHMV) (Vienna, Austria), Zoologische Staatssammlung München (ZSM) (Munich, Germany) and several private collectors (see Acknowledgements). To exclude the use of possibly shrunken museum material, almost all Plecoptera samples analysed in this study were loaned as alcohol specimens and dried at the critical point (EM CPD300, Leica Microsystems GmbH, Wetzlar, Germany) prior to scanning. Six airdried specimens from the Paris and Berlin collections, in which no shrinking could be detected, were used as well.

### (b) Tomography scanning and data processing

Head shape was investigated using synchrotron radiation micro-computed tomography (SR-µCT). This allowed for a high spatial resolution, high tissue contrast and rapid image acquisition. 177 specimens were scanned at the imaging cluster of the KIT Light Source (Karlsruhe Institute of Technology (KIT), Karlsruhe, Germany), 39 specimens at the IBL-P05 imaging beamline [40–42] (operated by the Helmholtz-Zentrum Hereon at PETRA III, Deutsches Elektronen Synchrotron (DESY), Hamburg, Germany), and one specimen at the TOMCAT beamline [43] (Swiss Light Source (SLS), Paul-Scherrer-Institute (PSI), Villigen, Switzerland). Two larger specimens were scanned with a commercially available µCT-machine (phoenix nanotom, General Electric, Boston, MA)

operated by Hereon. Rotated, three-dimensional regions of interest (ROIs) of the insect heads were manually extracted from the virtual image stacks and downsampled to less than 300 MB using a custom macro for FIJI [44] available online (github.com/Peter-T-Ruehr/stack_ cropping). A second Fiji macro (github.com/Peter-T-Ruehr/checkpoint_converter) was used to convert the image stacks to 'Checkpoint' files (*.ckpt) including their associated *.tif stacks to skip the manual import within 'Checkpoint' v. 6 (Stratovan Corporation, Davis, CA). Downsampled tomography scans are available at Zenodo (doi:10.5281/zenodo.4280412).

## (c) Head shape quantification with three-dimensional geometric morphometrics

Shape was quantified by using 3D geometric morphometrics (41 landmarks per species). Ten homologous type 1 landmarks, six type 2 landmarks [45] and 25 curve sliding semilandmarks (electronic supplementary material, figure S1a, tables S6 and S12) were placed on each 3D head model in Checkpoint to capture the head shape diversity of our sample. All type 2 landmarks and semilandmarks lie along the midsagittal plane of the head. Mandible shape was characterized by six type 1 landmarks (electronic supplementary material, figure S1b, tables S6 and S12). To skip manual landmark export in Checkpoint, a custom script (github.com/Peter-T-Ruehr/ checkpoint_importer_for_R) was used to import the landmarks of all species into the programming environment 'R' v. 4.0.5 [46] directly from the Checkpoint files. Generalized procrustes analyses (GPAs) were performed using 'gpagen' in 'geomorph' v. 3.3.1 [47] to remove the effects of non-shape variation from the dataset [48,49]. Sliding of the semilandmarks during the GPA was based on minimizing bending energy. Procrustes distance outliers were identified for each superfamily using the 'plotOutliers' function in geomorph and their landmarks were double checked. Head shape variation was subsequently investigated via phylogenetic PCAs (pPCAs [28,50]) and visualized with the 'phylomorphospace' function in 'phytools' v. 0.6.99 [51].

## (d) Database on ecological and morphological traits

We established a novel literature database to link life stage-specific ecological data with adult shape variation by screening 1950 literature sources, of which 960 were informative (listed in electronic supplementary material, table S11), for information on the following traits: 'microhabitat' and 'feeding habits' (for both nymphs and adults), and 'hydrodynamic pressure' (for nymphs). For each of these characters, we defined several character states (electronic supplementary material, table S7). Following Wilman et al. [52], who introduced a standardized interpretation of ecological and morphological wording, the often non-quantitative expert descriptions of traits were translated into semiquantitative information about the relative importance of this trait within its category in scores from 0 to 100% in 10% intervals. The standardized literature screening followed the same protocol for each species: a Google Scholar search with the full species name in quotes was queried in 'Publish or Perish' v. 7 [53]. The first 20 publications, sorted by 'rank', were checked for ecological information. In many cases, secondary literature based on the reference list of a given publication was searched. Additionally, we searched in all publications

listed in the respective 'species files' [38,39]. Literature data on species name synonyms, which were automatically retrieved from the Global Biodiversity Information Facility website (https://www.gbif.org) via the 'rgbif' package [54], was also searched for in the same way. Literature screening was finished on 3 June 2020, resulting in 3380 ecological data entries (electronic supplementary material, tables S1–S5). General statements about the ecology of taxon levels higher than species (such as genus, subfamily or family) were not added to the databank because even congeners may differ in their ecology. Early general contentions on non-feeding Plecoptera [55–59] were not taken into account because they have been widely disproven in later studies (see electronic supplementary material, tables S2 and S3). If information on different nymphal instars was available, only data on the last instar was taken, because we were interested in the effect of only the last (metamorphic) moult on adaptive decoupling.

## (e) Phylogenetic supertree generation

We digitized a Bayesian inference tree of Dermaptera based on five loci (18S rDNA, 28S rDNA, COI, Histone 3, and Tubulin Alpha I [60]) and a Bayesian inference tree of Plecoptera based on mitogenomes [61] using the 'phylo.tracer' function of the 'physketch' package v. 0.1 [62]. We manually added missing taxa to these phylogenies either by substituting closely related species or by using 'bind.tip' in phytools. Subsequently, we used the 'chronos' function in 'ape' v. 5.3 [63] to fit chronograms based on the branch lengths of the original phylogenies and the median node ages of the most recent common ancestors (MRCAs), according to Misof et al. [30], of Dermaptera (79.39 Ma) and Plecoptera (167.41 Ma). These phylogenies were then combined to a supertree using the median node age of the MRCA (302.05 Ma) [30] of Dermaptera and Plecoptera. Numerical imprecisions were eliminated by the 'force.ultrametric' function in phytools. The resulting ultrametric supertree (electronic supplementary material, figure S5) was pruned to only contain the taxa present in our analysis.

## (f) Models of shape evolution

Some of the functions used in this study require that the mode of shape evolution follows a Brownian motion model. To check this, we ran 'fitContinuous' in 'geiger' v. 2.0.6.2 [64] and tested which of the following models most closely describes the shape evolution in our dataset: 'BM', Brownian motion [65]; 'OU', Ornstein-Uhlenbeck [66]; 'EB', early burst [67], also known as 'ACDC' (accelerated/decelerated) [68]; 'lambda', phylogeny predicts covariance of shape among species [69]; and 'white noise', non-phylogenetic white-noise [64]. Using 'aic.w' in phytools, we compared the sample-size corrected Akaike information criterion (AICc) of the fitted models (electronic supplementary material, table S8) and found that the Brownian motion model had the best fit (AICc = −827.6, AICc-weight = 0.526). We thus concluded that the assumption of a Brownian motion model of shape evolution is sufficiently met by our dataset.

## (g) Allometric and phylogenetic signal

We analysed the effect of size on head shape in earwigs and stoneflies by performing a regression of the Procrustes-aligned shape data against log-centroid size using 'procD.lm' in geomorph with 10 000 permutations. We tested if earwigs and stoneflies

**Table 1.** Multivariate integrations of head shape (left) and mandible shape (right) of earwigs and stoneflies with ecological covariates expressed as effect sizes (z-scores) of phylogenetic partial least square results. Non-significant interactions left blank. See electronic supplementary material, table S9 for more statistical details and all test results. hydrodyn. p., hydrodynamic pressure; microh., microhabitat.

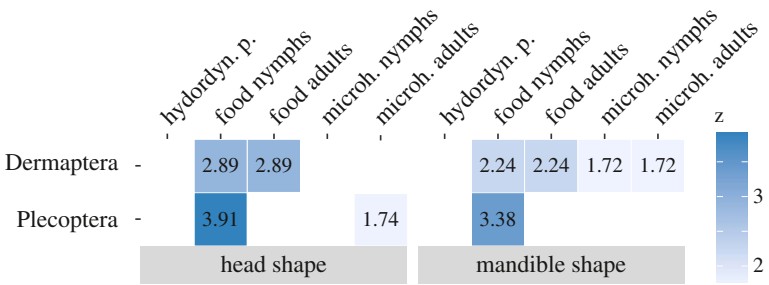

|  | \multicolumn{5}{c}{head shape} | \multicolumn{5}{c}{mandible shape} |
|---|---|---|---|---|---|---|---|---|---|---|
|  | hydrodyn. p. | food nymphs | food adults | microh. nymphs | microh. adults | hydrodyn. p. | food nymphs | food adults | microh. nymphs | microh. adults |
| Dermaptera | - | 2.89 | 2.89 |  |  | 2.24 | 2.24 | 1.72 | 1.72 |  |
| Plecoptera | - | 3.91 |  |  | 1.74 | 3.38 |  |  |  |  |

share common allometries by comparing a linear model with the null hypothesis of unique allometric slopes (shape ~ log(centroid size) × order) versus a linear model with the null hypothesis of a common allometric slope (shape ~ log(centroid size) + order). The same test was repeated for models with unique or common allometric slopes of superfamilies. All model fits were compared with an ANOVA. In downstream analyses, we accounted for the allometric effect using the residuals of a regression of shape on centroid size. The multivariate nature of the results of the principal coordinate analyses (PCoAs) of the ecological data (see below) did not allow for separate comparisons of allometric slopes of ecological groups.

Phylogenetic signal in the Procrustes-aligned shape data was evaluated by calculating the $K_{mult}$ statistic [70], a multivariate generalization of the $K$ statistic [68], using 'physignal' in geomorph with 10 000 iterations.

## (h) Integration of ecology and shape

To test our hypotheses that adult head and mandible shape in Dermaptera and Plecoptera covaries to varying degrees with ecological characteristics of either nymphs and/or adults, we calculated the degree of integration of the Procrustes shape data with the extracted multivariate ecological traits. We first calculated the Bray-Curtis dissimilarity index [71] of all species using 'vegdist' in 'vegan' v. 2.5-6 [72]. We ran PCoAs on these dissimilarity matrices using 'pcoa' in ape (electronic supplementary material, figure S4). The integration between shape and the PCoA vectors of the multivariate traits was separately calculated for each trait using 'phylo. integration' in 'geomorph' with 10 000 iterations (table 1; electronic supplementary material, table S9). This function identifies integration of multivariate traits while accounting for the phylogenetic non-independence of taxa by using an evolutionary covariance matrix under a Brownian motion model of evolution in the partial least squares (PLS) assessment of trait covariation [73]. Additionally, two-block PLS analyses were run to test for the integration of ecology and shape without considering phylogeny using 'two.b.pls' in geomorph (electronic supplementary material, table S9). Since we had to filter our data before the analyses according to species coverage for the ecological data, species numbers were different in every analysis for each ecological character. Additionally, the number of Procrustes coordinates in the shape data of head capsules versus mandibles on the one hand, and the number of PCos in the ecological data on the other hand, varied considerably. Both factors influence the results of the PLS correlation coefficient, because this coefficient is dependent on the number of

specimens and trait characters [74]. In order to be able to compare the explanatory values of the phylogenetic PLS analyses, we used 'compare.pls' in geomorph which calculates the effect sizes as z-scores. All above described analyses were carried out for the whole dataset (Dermaptera & Plecoptera) as well as for dermapteran and plecopteran subsets independently (electronic supplementary material, table S9).

All further downstream analyses were performed (i) not accounting for allometry or phylogeny, (ii) accounting for allometry only, (iii) accounting for phylogeny only and (iv) accounting for both allometry and phylogeny (electronic supplementary material, table S9). We only report the results of the phylogeny-corrected analyses in the main text.

## (i) Morphological disparity

Differences in morphospace occupation between Dermaptera and Plecoptera were estimated by running separate Procrustes-alignments of the shape data on the order-level. The morphological disparity for each of the order-subsets was calculated with 'morphol.disparity' in geomorph. Additionally, we compared the adult head shape disparity of perloidean stoneflies versus all other stoneflies and earwigs and non-perloidean stoneflies versus earwigs.

# 3. Results and discussion

## (a) Adult head shape evolution is allometrically and phylogenetically structured

Head size (measured as log (centroid size)) has a significant but weak influence on head shape (Procrustes ANOVA, $R^2 = 0.044$, $p = 1e^{-4}$, $n = 219$). The ANOVA comparing the log-transformed linear relationships of head size and shape in earwigs and stoneflies yielded a statistically significant difference of allometric slopes of the two orders (ANOVA, $R^2 = 0.026$, $p = 1e^{-4}$, $n = 219$). However, the low explanatory value of the model indicates low biological meaningfulness of this slope heterogeneity. ANOVA analyses on the allometric slopes of superfamily level taxonomic groups yielded similar results ($R^2 = 0.054$, $p = 4e^{-4}$). The inspection of the prediction lines [75] and regression scores [76] of the allometric analyses (electronic supplementary material, figure S3) showed that allometric slopes do not diverge greatly from each other in the different taxonomic groups. Given these results, we treated the whole dataset as if all taxa had a common allometric slope when correcting for allometric effects.

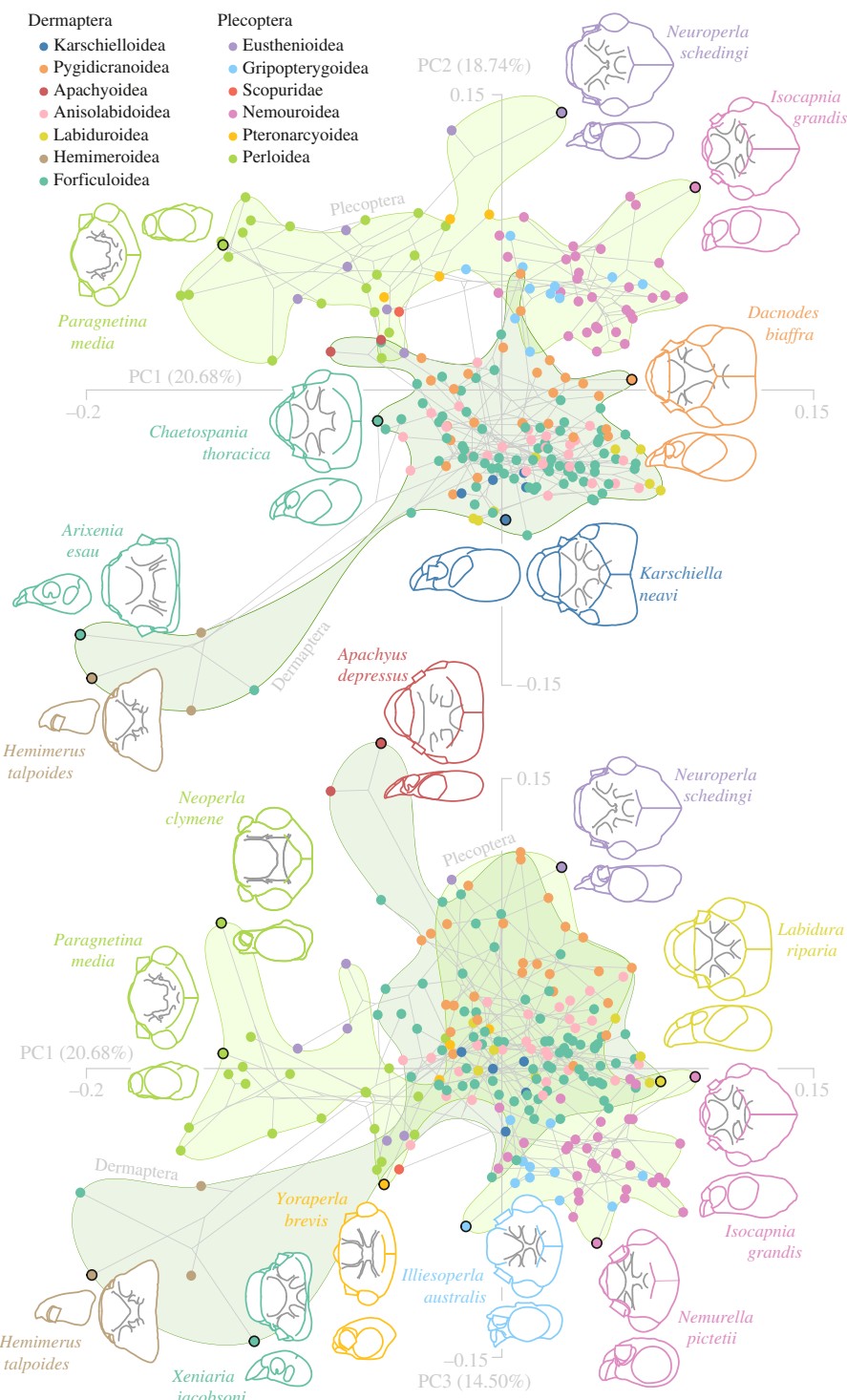

**Figure 2.** Head capsule phylomorphospace of adult earwigs and stoneflies. Principal component (PC) 1 and PC2 (upper part) and PC1 and PC3 (lower part) account for approximately 54% of the shape variation. Point colours represent superfamily level memberships, smoothed convex hulls show order-level memberships. Schematic drawings of the dorsal (including tentorial structures in grey) and lateral head shapes are based on µCT scans. Drawings are shown for selected species (black bordered points) at the edges of the phylomorphospace spanned by the first three PCs. Schematics not to scale. See electronic supplementary material, figure S3 for an overview of the first 8 PCs. (Online version in colour.)

The dataset also contains significant phylogenetic signal ($K_{mult} = 0.22$, $p = 1e^{-4}$, $n = 219$). $K_{mult}$ values below 1 indicate that the head shapes of closely related species are less similar to each other than expected under a Brownian motion model of evolution and could be explained by adaptive components in their evolution that do not follow the underlying phylogeny [68,70].

The results of the analyses with and without allometric corrections differ only slightly in their explanatory values and z-scores but not in the general pattern of influences of ecological factors on adult head and mandible shape. Not

correcting for phylogeny, however, resulted in much higher correlations of all ecological factors with shape (electronic supplementary material, table S9), indicating that closely related taxa generally share more common ecologies.

## (b) Convergent evolution into unique ecological niches

The first principal component (PC1) axis of an allometry- and phylogeny-corrected PC-analysis of head shape accounts for 20.68% of the variation and, generally, separates

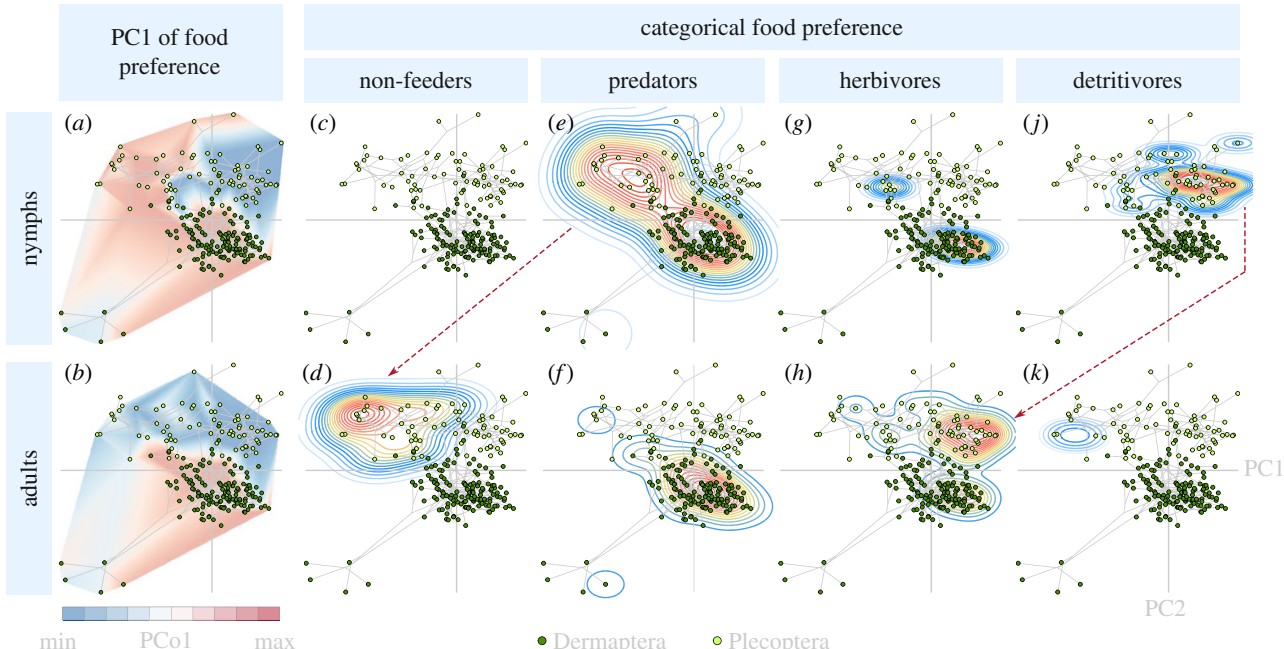

**Figure 3.** Morpho-ecological landscape illustrating the relationship of adult head shape with the feeding ecologies of nymphs (upper tiles) and adults (lower tiles). All tiles are overlaid with the phylomorphospace explained by the first two principal components of adult head shape in earwigs (dark green) and stoneflies (bright green). PC1 of nymphal food preference (a) mainly represents detritivoric (blue) versus predatory (red) habits, while PCo1 of adult feeding ecology (b) mainly represents herbivorous (blue) versus predatory (red) habits (see electronic supplementary material, figure S4 for PCoA biplots). Tiles (c–k) show density estimations of species with mainly non-feeding, (c,d), predatory (e,f), herbivorous (g,h) and (j,k) detritivoric nymphs and adults, respectively, to illustrate shifts of feeding habits across metamorphosis. Notable shifts in feeding habits are indicated by dashed arrows for clarity. (Online version in colour.)

short headed taxa (left side, figure 2) from taxa with more elongated head shapes (right side, figure 2). PC2 (18.74%) separates earwigs from stoneflies. Within earwigs, the phylomorphospace spanned by the first three PCs mainly separates the families Hemimeridae and Arixeniidae from all other lineages. These two families share a phylomorphospace region at the lower ends of PC1, PC2 and PC3 (14.5%). They do not form a monophyletic clade [60], but convergently evolved epizoic lifestyles and live, at least partly, in the fur of mammals [77–80]. All other earwig (super)families cluster near the centre of PCs1–3 (figure 2), a relative shape homogeneity which could be explained by the uniform feeding habits of these groups (see electronic supplementary material, tables S2 and S3 for taxon specific feeding habit extractions).

Within the stonefly morphospace, PC1 mainly describes differences between the species with predatory nymphs (most of them belonging to Perloidea and Eusthenioidea) and those with detritivoric nymphs (figures 2 and 3). Both Eusthenioidea and Perloidea occupy a similar phylomorphospace region at the upper region of PC1 and lower region of PC2, despite the fact that they are geographically and phylogenetically separated: Eusthenioidea, belonging to the suborder Antarctoperlaria, are restricted to the Southern Hemisphere, while Perloidea, belonging to the suborder Arctoperlaria, are, with a few secondary exceptions, restricted to the Northern Hemisphere [81]. Many lineages of Perloidea [82] and some lineages of Eusthenioidea [83,84] are able to fully develop their eggs within the last nymphal instar already. This results in a drastically reduced time to oviposition of a few days compared to many days or weeks in other species [82], and possibly lowered selection pressures on adult head shapes that are related to a regular uptake of nutrient-rich food [2].

## (c) Stronger ontogenetic niche shifts may result in lower adaptive decoupling in hemimetabolan insects

Multivariate, phylogenetically corrected integration tests of the Procrustes-aligned shape data against the results of the principal coordinate analyses (PCoA) of ecological covariates show that the head shape of adult stoneflies is most strongly influenced by the feeding habits of their nymphs ($R^2 = 0.79$, $z = 3.91$, $p = 1e^{-4}$, $n = 46$), and not by feeding habits of the adults themselves ($p = 0.41$, $n = 37$; table 1; electronic supplementary material, table S9). Nymphal feeding habits are also the only significant covariate of adult mandible shape in our analysis ($R^2 = 0.73$, $z = 3.38$, $p = 1e^{-4}$, $n = 47$). This high correlation of nymphal ecology and adult morphology indicates a low degree of adaptive decoupling: the last moult between the nymphal and adult stage in stoneflies does not seem to facilitate a disruption of trait coupling. Despite the strong ontogenetic niche shift resulting from the amphibiotic life style of stoneflies (figure 1; electronic supplementary material, tables S4 and S5), adult head and mandible morphology could not evolve independently from nymphal ecology. In addition to the habitat shift, changes in feeding ecology also occur across stonefly metamorphosis: predatory stonefly nymphs mostly metamorphose into liquid-feeding or non-feeding adults, while detritivoric stonefly nymphs mostly become herbivorous adults (figure 3; electronic supplementary material, tables S2 and S3). Selection pressures that act on the nymphal stage of stoneflies therefore seem to outweigh those acting on their adult stage, so that adult shape evolution is mainly driven by nymphal selection pressures.

Earwigs, on the other hand, show a relatively high correlation of both nymphal and adult feeding preference with adult head shape ($R^2 = 0.73$, $z = 2.89$, $p = 9e^{-4}$, $n = 35$;

table 1). They do not undergo major ontogenetic niche shifts when reaching adulthood, so their shape evolution may be equally adapted to the largely congruent selection pressures of both life phases. However, the degree of adaptive decoupling that could possibly be realized in earwigs may be as limited as in stoneflies since they also enter the adult phase with a single moult. Low degrees of adaptive decoupling in earwigs could therefore be either adaptively beneficial due to overlapping selection pressures, or the result of the limited possibility of metamorphic change across the final moult, or a combination of both effects.

## (d) Stronger ontogenetic niche shifts are accompanied by higher adult shape disparity

If patterns of variation are decoupled across metamorphosis, ecologically divergent life phases can effectively evolve towards unique selection pressures. Our data show that adult head shape is, however, not adaptively decoupled from nymphal ecology in earwigs and stoneflies. Instead, nymphal ecology drives adult head shape evolution (table 1). We hypothesized that this trait correlation would constrain the diversification of adult head shape, because head shape would not be free to evolve towards the unique selection pressures of the adult stage. Especially for stoneflies with strong ontogenetic niche shifts but high trait correlations, we expected that adult head shapes show a low shape disparity. Contrary to our expectations, however, adult head shape disparity in the amphibiotic stoneflies is significantly higher than in the fully terrestrial earwigs (Procrustes variance = 0.018 versus 0.013; $p = 3e^{-3}$). More detailed analyses revealed that highest levels of adult head shape disparity within stoneflies are concentrated at the superfamily Perloidea: when this group is excluded from the analysis, disparity within earwigs and stoneflies does not significantly differ from each other (0.013 versus 0.016; $p = 0.26$). Indeed, Perloidea alone show a significantly higher head shape disparity (0.026) than non-perloidean stoneflies ($p = 2e^{-3}$) and earwigs ($p = 3e^{-4}$). This observation could be explained by the fact that perloidean stoneflies do not rely on frequent feeding of hard food in the adult stage in order to sustain egg development, because nutrients have been already stored by the predatory nymphs [37,82]. The adaptive importance of feeding-related head structures in adult Perloidea might therefore be lowered, and the relative weight of selection pressures on the nymphal stage may be increased. This could have facilitated the evolution into new ecological niches in Perloidea, because adaptive conflicts between the life phases are reduced, possibly resulting in the observed higher head shape disparity of this group. Consequently, low degrees of adaptive decoupling can still facilitate increased phenotypic disparity.

## 4. Conclusion

About 80% of all animals show a complex life cycle with distinct life phases. Such life phases are characterized by phenotypic

adaptations to their phase-specific ecological niches. To avoid adaptive conflicts, traits can be decoupled between life phases, and metamorphosis is thought to aid in the breakup of trait correlations. We showed that adult head shape evolution in earwigs and stoneflies, two closely related hemimetabolous insect taxa, is not decoupled from juvenile ecology despite sometimes strong ontogenetic niche shifts. We therefore conclude that the hemimetabolan metamorphosis in earwigs and stoneflies does not facilitate a disruption of trait couplings, resulting in a constrained phenotypic evolution of the adult phase. Additionally, stronger food-related ontogenetic niche shifts within stoneflies have resulted in higher shape disparity in the adults of some stonefly families, possibly because of the liberation of the adult stage from food-related functions.

Data accessibility. All 219 cropped and down-sampled µCT scans used in this study have been deposited at Zenodo.org (https://doi.org/10.5281/zenodo.4280412). Full datasets are available upon request from the corresponding author. The R code used in this study is publicly available on GitHub (https://github.com/Peter-T-Ruehr/Adaptive_decoupling_insects).

The data are provided in electronic supplementary material [85].

Authors' contributions. P.T.R.: conceptualization, data curation, formal analysis, investigation, methodology, project administration, visualization, writing-original draft, writing-review and editing; T.v.d.K.: investigation, methodology, resources, software, writing-review and editing; T.F.: investigation, methodology, resources, software, writing-review and editing; J.U.H.: investigation, methodology, resources, software, writing-review and editing; F.W.: investigation, methodology, resources, software, writing-review and editing; E.B.: investigation, methodology, resources, software, writing-review and editing; C.E.: investigation, writing-review and editing; M.F.: investigation, writing-review and editing; T.B.: resources, writing-review and editing; A.B.: conceptualization, formal analysis, funding acquisition, investigation, project administration, resources, supervision, writing-review and editing.

All authors gave final approval for publication and agreed to be held accountable for the work performed therein.

Competing interests. We declare we have no competing interests.

Funding. P.T.R., M.F. and A.B. were supported by the European Research Council (ERC) under the European Union's Horizon 2020 research and innovation program (grant agreement no. 754290, 'Mech-Evo-Insect'). A.B. and C.E. were supported by the Deutsche Forschungsgemeinschaft (DFG) under the Individual Research Grants program (grant agreement no. BL 1355/4-1). µCT-scanning was funded by the following, facility-specific grants awarded to A.B. Hereon at DESY: I-20170190, I-20170896, I-20190019, SLS: 20171469.

Acknowledgements. B. Price (NHMUK), H. Zettel, D. Zimmermann (both NHMV), J. Deckert (MfN), R. Peters (ZFMK) and L. Hendrich (ZSM) kindly organized access to and loan of specimens. O. Béthoux, A. Caires, S. Mtow, P. Pessacq and P. Zwick generously provided specimens from private collections. We acknowledge the KIT light source for provision of instruments at their beamlines and we would like to thank the Institute for Beam Physics and Technology (IBPT) for the operation of the storage ring, the Karlsruhe Research Accelerator (KARA). M. Dabringhaus, A. Huenebeck, U. Kolesnikov, M. Prelle, L. Scheele, I. Javed and C. Voss are thanked for their help with µCT data processing. The infrastructural support of the Büschges group (University of Cologne) during early parts of the study is highly appreciated. We are grateful for helpful comments by X. Belles (CSIC-Universitat Pompeu Fabra), J. Collet (Centre national de la recherche scientifique), S. Fellous (Université de Montpellier) and J. W. Truman (University of Washington) on an earlier version of this manuscript. We thank two anonymous reviewers for their helpful comments.

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
