## [Peer Review File · Proceedings of the Royal Society B: Biological Sciences]

Review History

RSPB-2021-0616.R0 (Original submission)

Review form: Reviewer 1

Recommendation

Accept with minor revision (please list in comments)

Scientific importance: Is the manuscript an original and important contribution to its field?

Excellent

General interest: Is the paper of sufficient general interest?

Excellent

Quality of the paper: Is the overall quality of the paper suitable?

Good

Is the length of the paper justified?

Yes

Should the paper be seen by a specialist statistical reviewer?

Yes

Do you have any concerns about statistical analyses in this paper? If so, please specify them explicitly in your report.

No

It is a condition of publication that authors make their supporting data, code and materials available - either as supplementary material or hosted in an external repository. Please rate, if applicable, the supporting data on the following criteria.

Is it accessible?

Yes

Is it clear?

Yes

Is it adequate?

Yes

Do you have any ethical concerns with this paper?

No

Comments to the Author

The Plecoptera (stoneflies) and Dermaptera (earwigs) are sister insect orders within the Polyneoptera, both hemimetabolans, then. However, they have a very different life cycle: stoneflies have a juvenile aquatic life whereas the adults are terrestrial, while in earwigs all life is terrestrial. Two well-chosen groups, then, to study whether the change from aquatic to terrestrial might determine a morphological change in hemimetabolans. The authors address this question through an extensive study that includes 144 earwig species and 75 stoneflies, a total of 219 species covering all extant families of both orders, which is a very robust sample. As was quite expected in hemimetabolan insects, there are no major morphological changes (represented in this case by the characteristics of the cephalic capsule) in the transition from nymphs to adult, even in stoneflies that transit from aquatic to terrestrial. However, the authors observe some divergence between nymphs and adults in some perloids, within the plecopterans, which can lead to interesting conclusions, which we will comment on later.

The work is rigorous, with a very robust sample, as mentioned above, and very exhaustive analyses. I must confess, however, that I am not familiar with most analytical and statistical approaches, and cannot comment in detail on these matters. Regarding conceptual aspects, I do not detect any problems worthy of mention. For all these reasons, my comments will be limited almost exclusively to a particularly formal aspect, as I detail below.

Specific comments

1. Title. "Juvenile ecology drives adult morphology in two insect lineages". The more precise word "orders", instead of "lineages", could be used.
2. L. 30. "incomplete metamorphosis". There is a scientific term for that: "hemimetabolan metamorphosis". Colloquially, better using "direct metamorphosis".
3. L. 31-32. "that high phenotypic disparity can be realised even when distinct life phases are prevented from evolving independently". Not clear, especially the expression "prevented from evolving independently".
4. L: 50. "Winged insects". "Winged" unnecessary.
5. L: 52. "strong anatomical remodelling". Rather "strong morphological remodelling", or even "morphoanatomical".
6. L: 57. "an intermittent pupal stage". "intermittent"? Do you mean "intermediate"? You can simply delete "intermittent".
7. L. 58. "dragonflies, mayflies and stoneflies". Perhaps add the Latin name of these orders: "dragonflies (Odonata), mayflies (Ephemeroptera) and stoneflies (Plecoptera)", to be more precise.
8. L. 59. "stonefly taxa". Read "stonefly species".

9. L. 61-62. "incomplete metamorphosis". Better "direct metamorphosis".
10. L. 72-74. "We chose to study the two closely related [26,27] hemimetabolous insect taxa earwigs (Polyneoptera: Dermaptera, ~2000 described species[28]) and stoneflies (Polyneoptera: Plecoptera, ~3400 species[28])". Not only "closely related". They are sister orders, within the Polyneoptera (Misof et al., 2014; Wang et al., 2016). This sister-group relationship is an important requisite for evolutionary comparisons. In addition, please refer to "orders" not "taxa".
11. L. 75-76. "a low performance flight apparatus". Much poorer in dermapterans by far.
12. L. 91. "'Muséum national d'Histoire Naturelle". Read "Muséum national d'Histoire naturelle".
13. L. 220. "They are not closely related to each other". The expression "closely related" sounds ambiguous.
14. L. 244. "Stronger ontogenetic niche shifts result in lower adaptive decoupling" Not sure that the first proposition brings the second. I rather thin on something like "Strong ontogenetic niche shifts does not involve significant adaptive decoupling"
15. L. 314-315. "incomplete metamorphosis". Better "direct metamorphosis".
16. L. 363-364. "Belles X. 2020 Insect Metamorphosis. London, San Diego, Cambridge, Oxford: Elsevier Academic Press. (doi:10.1016/C2016-0-04530-8)". The exact reference is: . "Belles X. 2020 Insect Metamorphosis. From natural history to regulation of development and evolution. London: Academic Press. (doi:10.1016/C2016-0-04530-8)".
17. Pages 24-102. What is the Legend of this long table?

Review form: Reviewer 2

Recommendation

Accept with minor revision (please list in comments)

Scientific importance: Is the manuscript an original and important contribution to its field?

Excellent

General interest: Is the paper of sufficient general interest?

Excellent

Quality of the paper: Is the overall quality of the paper suitable?

Excellent

Is the length of the paper justified?

Yes

Should the paper be seen by a specialist statistical reviewer?

No

Do you have any concerns about statistical analyses in this paper? If so, please specify them explicitly in your report.

No

It is a condition of publication that authors make their supporting data, code and materials available - either as supplementary material or hosted in an external repository. Please rate, if applicable, the supporting data on the following criteria.

Is it accessible?

Yes

Is it clear?

Yes

Is it adequate?

Yes

Do you have any ethical concerns with this paper?

No

Comments to the Author

I have reviewed the manuscript entitled “Juvenile ecology drives adult morphology in two insect lineages”, in which the authors elegantly demonstrate the influence that a complex life cycle and ontogenetic niche shift has on two contrasting insect groups. The study uses 3D imaging via synchrotron Computed Tomography to produce an impressive dataset demonstrating the morphological diversity at two age classes in each group, and they compile an impressive and detailed dataset of diets in order to test hypotheses regarding the ecological influence on disparity.

Put simply, I love this paper. It was a pleasure to read. The figures are excellent, and the dataset impressive, setting the bar high for macroevolutionary studies to come.

I have only a few stylistic comments:

Lines 65-67, suggest writing “that” before the first parentheses to avoid unnecessary duplication

Line 99 – spell out uCT first, and micro-computed tomography.

Line 108- checkpoint file? Do you mean you wrote custom .ckpt files in order to open them easily in checkpoint for landmarking. That’s neat. Please say .ckpt in parentheses after checkpoint.

Line 111 - it’s not customary to total up the landmarks for all specimens. Instead give the number of landmarks per specimen (41 I believe).

Lines 171 -177 – Need to be explicit in how allometry was removed (assume you are using the residuals). But now was phylogeny removed? Also needs more explicit methods here.

Furthermore, it is important to show that the residuals approach can be used here – if the slopes of your groups differ then a single linear regression model for all data will not be accurate.

Line 245- needs hyphen between phylogenetically and corrected.

Figure 2 caption – please be more specific about the “at extremes of pc space”. I believe you are plotting the min and max values of the PC scores on each axis, when all other axes have a score of 0 (the standard way), but this is different from what can be interpreted as the extremes of this space.

Decision letter (RSPB-2021-0616.R0)

09-Apr-2021

Dear Mr Rühr:

Your manuscript has now been peer reviewed and the reviews have been assessed by an Associate Editor. The reviewers’ comments (not including confidential comments to the Editor) and the comments from the Associate Editor are included at the end of this email for your reference. As you will see, the reviewers and the Editors have raised some concerns with your manuscript and we would like to invite you to revise your manuscript to address them.

We do not allow multiple rounds of revision so we urge you to make every effort to fully address all of the comments at this stage. If deemed necessary by the Associate Editor, your manuscript will be sent back to one or more of the original reviewers for assessment. If the original reviewers

are not available we may invite new reviewers. Please note that we cannot guarantee eventual acceptance of your manuscript at this stage.

Research ethics:

Use of animals and field studies:

It is a condition of publication that you make available the data and research materials supporting the results in the article. Please see our Data Sharing Policies (<https://royalsociety.org/journals/authors/author-guidelines/#data>). Datasets should be deposited in an appropriate publicly available repository and details of the associated accession number, link or DOI to the datasets must be included in the Data Accessibility section of the article (<https://royalsociety.org/journals/ethics-policies/data-sharing-mining/>). Reference(s) to datasets should also be included in the reference list of the article with DOIs (where available).

Please submit a copy of your revised paper within three weeks. If we do not hear from you within this time your manuscript will be rejected. If you are unable to meet this deadline please let us know as soon as possible, as we may be able to grant a short extension.

Best wishes,
Dr Sasha Dall
mailto:proceedingsb@royalsociety.org

Associate Editor
Board Member: 1
Comments to Author:

Thank you for the opportunity to review this manuscript, which tests the hypothesis of adaptive decoupling of life phases across metamorphosis with a novel analysis of a large dataset from hemimetabolous insect taxa. Reviews from two referees have now been received, both of which were very positive and noted a high likelihood for strong interest from the audience of Proceedings B.

Both referees made suggestions to help clarify points in the MS. Some specific recommended choices of wording might be discretionary, but other recommendations include providing more detail on some of the methods applied. For example, Referee 2 particularly noted a need for additional detail in how allometry and phylogeny were considered in the analyses described in lines 171-177. In addition to the suggestions of the Referees, I would add a few further recommended corrections that I list below.

Thank you once again for your submission. I hope you find the referee comments to provide constructive guidance for revising your report on your study.

L19. Add a comma before "but".

L21. Change "niche, a process called" to "niche. This process is called..."

L44. Remove "e.g."

L45. Add "and" before "salamanders".

L48. "extend" should be "extent".

L68. Remove "the" after "in".

L208. Substitute "small" in place of "low".

L297-8. For "disparity between earwigs and stoneflies does not significantly differ," should "between" be "within" - because the comparison is between the disparity levels found within each lineage?

L317. Change "has" to "have".

Reviewer(s)' Comments to Author:

Referee: 1

Comments to the Author(s)

The Plecoptera (stoneflies) and Dermaptera (earwigs) are sister insect orders within the Polyneoptera, both hemimetabolans, then. However, they have a very different life cycle: stoneflies have a juvenile aquatic life whereas the adults are terrestrial, while in earwigs all life is terrestrial. Two well-chosen groups, then, to study whether the change from aquatic to terrestrial might determine a morphological change in hemimetabolans. The authors address this question through an extensive study that includes 144 earwig species and 75 stoneflies, a total of 219 species covering all extant families of both orders, which is a very robust sample. As was quite expected in hemimetabolans, there are no major morphological changes (represented in this case by the characteristics of the cephalic capsule) in the transition from nymphs to adult, even in stoneflies that transit from aquatic to terrestrial. However, the authors observe some divergence between nymphs and adults in some perloids, within the plecopterans, which can lead to interesting conclusions, which we will comment on later.

The work is rigorous, with a very robust sample, as mentioned above, and very exhaustive analyses. I must confess, however, that I am not familiar with most analytical and statistical approaches, and cannot comment in detail on these matters. Regarding conceptual aspects, I do not detect any problems worthy of mention. For all these reasons, my comments will be limited almost exclusively to a particularly formal aspect, as I detail below.

Specific comments

1. Title. "Juvenile ecology drives adult morphology in two insect lineages". The more precise word "orders", instead of "lineages", could be used.
2. L. 30. "incomplete metamorphosis". There is a scientific term for that: "hemimetabolans metamorphosis". Colloquially, better using "direct metamorphosis".
3. L. 31-32. "that high phenotypic disparity can be realised even when distinct life phases are prevented from evolving independently". Not clear, especially the expression "prevented from evolving independently".
4. L. 50. "Winged insects". "Winged" unnecessary.
5. L. 52. "strong anatomical remodelling". Rather "strong morphological remodelling", or even "morphoanatomical".
6. L. 57. "an intermittent pupal stage". "intermittent"? Do you mean "intermediate"? You can simply delete "intermittent".
7. L. 58. "dragonflies, mayflies and stoneflies". Perhaps add the Latin name of these orders: "dragonflies (Odonata), mayflies (Ephemeroptera) and stoneflies (Plecoptera)", to be more precise.
8. L. 59. "stonefly taxa". Read "stonefly species".
9. L. 61-62. "incomplete metamorphosis". Better "direct metamorphosis".
10. L. 72-74. "We chose to study the two closely related [26,27] hemimetabolous insect taxa earwigs (Polyneoptera: Dermaptera, ~2000 described species[28]) and stoneflies (Polyneoptera: Plecoptera, ~3400 species[28])". Not only "closely related". They are sister orders, within the Polyneoptera (Misof et al., 2014; Wang et al., 2016). This sister-group relationship is an important requisite for evolutionary comparisons. In addition, please refer to "orders" not "taxa".
11. L. 75-76. "a low performance flight apparatus". Much poorer in dermapterans by far.
12. L. 91. "Muséum national d'Histoire Naturelle". Read "Muséum national d'Histoire naturelle".
13. L. 220. "They are not closely related to each other". The expression "closely related" sounds ambiguous.
14. L. 244. "Stronger ontogenetic niche shifts result in lower adaptive decoupling" Not sure that the first proposition brings the second. I rather thin on something like "Strong ontogenetic niche shifts does not involve significant adaptive decoupling"
15. L. 314-315. "incomplete metamorphosis". Better "direct metamorphosis".

16. L. 363-364. "Belles X. 2020 Insect Metamorphosis. London, San Diego, Cambridge, Oxford: Elsevier Academic Press. (doi:10.1016/C2016-0-04530-8)". The exact reference is: . "Belles X. 2020

Insect Metamorphosis. From natural history to regulation of development and evolution. London: Academic Press. (doi:10.1016/C2016-0-04530-8)".
17. Pages 24-102. What is the Legend of this long table?

Referee: 2

Comments to the Author(s)

I have reviewed the manuscript entitled "Juvenile ecology drives adult morphology in two insect lineages", in which the authors elegantly demonstrate the influence that a complex life cycle and ontogenetic niche shift has on two contrasting insect groups. The study uses 3D imaging via synchrotron Computed Tomography to produce an impressive dataset demonstrating the morphological diversity at two age classes in each group, and they compile an impressive and detailed dataset of diets in order to test hypotheses regarding the ecological influence on disparity.

Put simply, I love this paper. It was a pleasure to read. The figures are excellent, and the dataset impressive, setting the bar high for macroevolutionary studies to come.

I have only a few stylistic comments:

Lines 65-67, suggest writing "that" before the first parentheses to avoid unnecessary duplication
Line 99 – spell out uCT first, and micro-computed tomography.

Line 108- checkpoint file? Do you mean you wrote custom .ckpt files in order to open them easily in checkpoint for landmarking. That's neat. Please say .ckpt in parentheses after checkpoint.

Line 111 - it's not customary to total up the landmarks for all specimens. Instead give the number of landmarks per specimen (41 I believe).

Lines 171 -177 – Need to be explicit in how allometry was removed (assume you are using the residuals). But now was phylogeny removed? Also needs more explicit methods here.

Furthermore, it is important to show that the residuals approach can be used here – if the slopes of your groups differ then a single linear regression model for all data will not be accurate.

Line 245- needs hyphen between phylogenetically and corrected.

Figure 2 caption – please be more specific about the "at extremes of pc space". I believe you are plotting the min and max values of the PC scores on each axis, when all other axes have a score of 0 (the standard way), but this is different from what can be interpreted as the extremes of this space.

Author's Response to Decision Letter for (RSPB-2021-0616.R0)

See Appendix A.

RSPB-2021-0616.R1 (Revision)

Review form: Reviewer 1

Recommendation

Accept as is

Scientific importance: Is the manuscript an original and important contribution to its field?
Excellent

General interest: Is the paper of sufficient general interest?

Excellent

Quality of the paper: Is the overall quality of the paper suitable?

Excellent

Is the length of the paper justified?

Yes

Should the paper be seen by a specialist statistical reviewer?

No

Do you have any concerns about statistical analyses in this paper? If so, please specify them explicitly in your report.

No

It is a condition of publication that authors make their supporting data, code and materials available - either as supplementary material or hosted in an external repository. Please rate, if applicable, the supporting data on the following criteria.

Is it accessible?

Yes

Is it clear?

Yes

Is it adequate?

Yes

Do you have any ethical concerns with this paper?

No

Comments to the Author

Thank you for providing the extra details for the methods. I am satisfied that this has been done appropriately.

Review form: Reviewer 2

Recommendation

Accept as is

Scientific importance: Is the manuscript an original and important contribution to its field?

Excellent

General interest: Is the paper of sufficient general interest?

Good

Quality of the paper: Is the overall quality of the paper suitable?

Excellent

Is the length of the paper justified?

Yes

Should the paper be seen by a specialist statistical reviewer?

No

Do you have any concerns about statistical analyses in this paper? If so, please specify them explicitly in your report.

No

It is a condition of publication that authors make their supporting data, code and materials available - either as supplementary material or hosted in an external repository. Please rate, if applicable, the supporting data on the following criteria.

Is it accessible?

Yes

Is it clear?

Yes

Is it adequate?

Yes

Do you have any ethical concerns with this paper?

No

Comments to the Author

The new version of the manuscript has been clearly improved with respect to the previous one. The authors have addressed all my comments satisfactorily, and my opinion is that the manuscript might be published as it is now.

Decision letter (RSPB-2021-0616.R1)

24-May-2021

Dear Mr Rühr

I am pleased to inform you that your manuscript entitled "Juvenile ecology drives adult morphology in two insect orders" has been accepted for publication in Proceedings B.

Data Accessibility section

Open Access

You are invited to opt for Open Access, making your freely available to all as soon as it is ready for publication under a CCBY licence. Our article processing charge for Open Access is £1700. Corresponding authors from member institutions (<http://royalsocietypublishing.org/site/librarians/allmembers.xhtml>) receive a 25% discount to these charges. For more information please visit <http://royalsocietypublishing.org/open-access>.

Your article has been estimated as being 9 pages long. Our Production Office will be able to confirm the exact length at proof stage.

Paper charges

Sincerely,
Dr Sasha Dall
Editor, Proceedings B
<mailto:proceedingsb@royalsociety.org>

Associate Editor:

Board Member: 1

Comments to Author:

Thank you for submitting the revised version of your manuscript, which has thoroughly addressed the comments raised by the referees. Your contribution to Proceedings B is appreciated.

Appendix A

Dear Editor,

we thank you and the two anonymous reviewers for the constructive comments, which helped us to improve the manuscript. We address and discuss all points below and added the line numbers in brackets to each comment in case they changed compared to the old manuscript. All suggested changes to wording and punctuation have been incorporated.

Sincerely yours,

Peter T. Rühr (on behalf of all co-authors)

Associate Editor

Board Member: 1

Comments to Author:

Thank you for the opportunity to review this manuscript, which tests the hypothesis of adaptive decoupling of life phases across metamorphosis with a novel analysis of a large dataset from hemimetabolous insect taxa. Reviews from two referees have now been received, both of which were very positive and noted a high likelihood for strong interest from the audience of Proceedings B.

Both referees made suggestions to help clarify points in the MS. Some specific recommended choices of wording might be discretionary, but other recommendations include providing more detail on some of the methods applied. For example, Referee 2 particularly noted a need for additional detail in how allometry and phylogeny were considered in the analyses described in lines 171-177. In addition to the suggestions of the Referees, I would add a few further recommended corrections that I list below.

Thank you once again for your submission. I hope you find the referee comments to provide constructive guidance for revising your report on your study.

L19. Add a comma before “but”.

← done.

L21. Change “niche, a process called” to “niche. This process is called...”

← done.

L44. Remove “e.g.”

← done.

L45. Add “and” before “salamanders”.

← done. (L.44)

L48. “extend” should be “extent”.

← done.

L68. Remove “the” after “in”.

← done.

L208. Substitute “small” in place of “low”.

← we changed it to “weak” which is more commonly used in literature. (L.232)

L297-8. For “disparity between earwigs and stoneflies does not significantly differ,” should “between” be “within” – because the comparison is between the disparity levels found within each lineage?

← yes, that is correct and we made the suggested changes. (L. 336)

L317. Change “has” to “have”.

← done. (L. 356)

Reviewer(s)' Comments to Author:

Referee: 1

The Plecoptera (stoneflies) and Dermaptera (earwigs) are sister insect orders within the Polyneoptera, both hemimetabolans, then. However, they have a very different life cycle: stoneflies have a juvenile aquatic life whereas the adults are terrestrial, while in earwigs all life is terrestrial. Two well-chosen groups, then, to study whether the change from aquatic to terrestrial might determine a morphological change in hemimetabolans. The authors address this question through an extensive study that includes 144 earwig species and 75 stoneflies, a total of 219 species covering all extant families of both orders, which is a very robust sample. As was quite expected in hemimetabolans, there are no major morphological changes (represented in this case by the characteristics of the cephalic capsule) in the transition from nymphs to adult, even in stoneflies that transit from aquatic to terrestrial. However, the authors observe some divergence between nymphs and adults in some perloids, within the plecopterans, which can lead to interesting conclusions, which we will comment on later.

The work is rigorous, with a very robust sample, as mentioned above, and very exhaustive analyses. I must confess, however, that I am not familiar with most analytical and statistical approaches, and cannot comment in detail on these matters. Regarding conceptual aspects, I do not detect any problems worthy of mention. For all these reasons, my comments will be limited almost exclusively to a particularly formal aspect, as I detail below.

Specific comments

1. Title. “Juvenile ecology drives adult morphology in two insect lineages”. The more precise word “orders”, instead of “lineages”, could be used.

← done.

2. L. 30. “incomplete metamorphosis”. There is a scientific term for that: “hemimetabolans metamorphosis”. Colloquially, better using “direct metamorphosis”.

← We corrected this to “hemimetabolans” as suggested. (L. 29)

3. L. 31-32. "that high phenotypic disparity can be realised even when distinct life phases are prevented from evolving independently". Not clear, especially the expression "prevented from evolving independently".

← Yes, we changed the ending of the sentence to "that high phenotypic disparity can even be realised when the evolution of distinct life phases is coupled."

4. L: 50. "Winged insects". "Winged" unnecessary.

← we used "winged" here to exclude the apterygote insects Archaeognatha ("jumping bristletails") and Zygentoma ("silverfish") that follow an ametabolan mode of development. Since the term "Insecta" in its strict sense is a synonym of Ectognatha and thus includes Archaeognatha and Zygentoma, we think that the use of "winged" clarifies the taxonomic orders we are referring to in this statement. (L. 49)

5. L: 52. "strong anatomical remodelling". Rather "strong morphological remodelling", or even "morphoanatomical".

← changed to "morphological remodelling". (L. 51)

6. L: 57. "an intermittent pupal stage". "intermittent"? Do you mean "intermediate"? You can simply delete "intermittent".

← agreed. We deleted "intermittent". (L 56)

7. L. 58. "dragonflies, mayflies and stoneflies". Perhaps add the Latin name of these orders: "dragonflies (Odonata), mayflies (Ephemeroptera) and stoneflies (Plecoptera)", to be more precise.

← agreed and changed as suggested. (L. 57f)

8. L. 59. "stonefly taxa". Read "stonefly species".

← done.

9. L. 61-62. "incomplete metamorphosis". Better "direct metamorphosis".

← We changed this, as suggested above, to "hemimetabolan metamorphosis", because "direct metamorphosis" could also point to the ametabolan mode of development of Archaeognath and Zygentoma.

10. L. 72-74. "We chose to study the two closely related [26,27] hemimetabolous insect taxa earwigs (Polyneoptera: Dermaptera, ~2000 described species[28]) and stoneflies (Polyneoptera: Plecoptera, ~3400 species[28])". Not only "closely related". They are sister orders, within the Polyneoptera (Misof et al., 2014; Wang et al., 2016). This sister-group relationship is an important requisite for evolutionary comparisons. In addition, please refer to "orders" not "taxa".

← According to Wang et al. 2016 (SciRep 6; 10.1038/srep38939), Dermaptera and Plecoptera indeed form a sister taxon "Dermoplecopterida" based on nuclear and mitochondrial genes. However, according to the comprehensive, transcriptome-based phylogenetic reconstructions of Misof et al. 2014 (Science 346; 10.1126/science.1257570) and Wipfler et al. 2019 (PNAS 166; 10.1073/pnas.1817794116), earwigs and stoneflies are not sister taxa. Here, Dermaptera form a monophyletic clade with the Zoraptera. Together, they form a sister relationship with the rest of the Polyneoptera, of which stoneflies are the earliest branching clade. Since transcriptome-based phylogenomics are usually more robust for the reconstruction of divergences of old taxa (the MRCA

of Dermaptera and Plecoptera lived ~300 mya), we prefer to follow the phylogenetic topology of Misof et al. and Wipfler et al. and did not change the term “closely related” in the manuscript. However, we changed “taxa” to “orders”, as suggested. (L. 73)

11. L. 75-76. “a low performance flight apparatus”. Much poorer in dermapterans by far.

← generally true, but there a quite a few earwigs that are potent and common flyers, and many stoneflies that are wingless or clumsy and even reluctant fliers, preferring to flee running on the ground instead of flying away.

12. L. 91. “Muséum national d'Histoire Naturelle”. Read “Muséum national d'Histoire naturelle”.

← done. (L. 92)

13. L. 220. “They are not closely related to each other”. The expression “closely related” sounds ambiguous.

← yes. We changed the sentence to “They do not form a monophyletic clade...” (L. 259)

14. L. 244. “Stronger ontogenetic niche shifts result in lower adaptive decoupling” Not sure that the first proposition brings the second. I rather thin on something like “Strong ontogenetic niche shifts does not involve significant adaptive decoupling”

← we agree that this is a bold statement in this form. We changed it to “Stronger ontogenetic niche shifts may result in lower adaptive decoupling in hemimetabolan insects” (L. 283)

15. L. 314-315. “incomplete metamorphosis”. Better “direct metamorphosis”.

← We changed this to “hemimetabolan metamorphosis” (s. above). (L. 354)

16. L. 363-364. “Belles X. 2020 Insect Metamorphosis. London, San Diego, Cambridge, Oxford: Elsevier Academic Press. (doi:10.1016/C2016-0-04530-8)”. The exact reference is: . “Belles X. 2020 Insect Metamorphosis. From natural history to regulation of development and evolution. London: Academic Press. (doi:10.1016/C2016-0-04530-8)”.

← done. (L. 404 f.)

17. Pages 24-102. What is the Legend of this long table?

← This table was uploaded as an Excel file during submission and automatically converted and added to the manuscript. The legend is entered into the resubmission mask as follows:

“Head capsule and mandible landmark coordinates of the 219 earwig and stonefly species analysed in this study.”

Reviewer(s)' Comments to Author:

Referee: 2

I have reviewed the manuscript entitled “Juvenile ecology drives adult morphology in two insect lineages”, in which the authors elegantly demonstrate the influence that a complex life cycle and ontogenetic niche shift has on two contrasting insect groups. The study uses 3D imaging via synchrotron Computed Tomography to produce an impressive dataset demonstrating the morphological diversity at two age classes in each group, and they compile an impressive and detailed dataset of diets in order to test hypotheses regarding the ecological influence on disparity.

Put simply, I love this paper. It was a pleasure to read. The figures are excellent, and the dataset impressive, setting the bar high for macroevolutionary studies to come.

I have only a few stylistic comments:

Lines 65-67, suggest writing “that” before the first parentheses to avoid unnecessary duplication

← done.

Line 99 – spell out uCT first, and micro-computed tomography.

← done.

Line 108- checkpoint file? Do you mean you wrote custom .ckpt files in order to open them easily in checkpoint for landmarking. That’s neat. Please say .ckpt in parentheses after checkpoint.

← done. We also added links to the GitHub repositories with the ImageJ macros for ROI definition and image stack import into Checkpoint (L. 108 ff.), and the R-script of direct import of Checkpoint landmark data into R. (L 120 ff.)

Line 111 - it’s not customary to total up the landmarks for all specimens. Instead give the number of landmarks per specimen (41 I believe).

← done. (L. 115)

Lines 171 -177 – Need to be explicit in how allometry was removed (assume you are using the residuals). But now was phylogeny removed? Also needs more explicit methods here. Furthermore, it is important to show that the residuals approach can be used here – if the slopes of your groups differ then a single linear regression model for all data will not be accurate.

← We have calculated and compared allometric slopes of taxonomic groups (orders and superfamilies) and found that models with heterogeneous slopes had better fits than models with a common slope. Yet, the explanatory power of the model fits with unique slopes ($R^2 = 0.026$ for orders and $R^2 = 0.054$ for superfamilies) was low. That is why we decided to continue to work with the allometry-corrected data. The difference between analyses with and without allometric corrections are slight, and the general pattern of ecological correlations does not change. We hope that by making our decision more transparent (s. new paragraphs in lines pointed to below, Supplementary Fig S2 and a new Supplementary Table S9), we satisfy the reviewer's rightful critique on this topic.

L. 182 ff. (Material and Methods)

L. 217 ff. (Material and Methods)

L. 233 ff. (Results and Discussion)

The phylogenetic correction of morphological integration with ecological traits is done within the 'geomorph' function 'phylo.integration'. We have not described this in detail, because it is explained in referenced publication: Adams & Felice (PLOS ONE 9; 10.1371/journal.pone.0094335). We added a short explanation to the text in L. 204 f.

Line 245- needs hyphen between phylogenetically and corrected.

← done. (L. 285)

Figure 2 caption – please be more specific about the “at extremes of pc space”. I believe you are plotting the min and max values of the PC scores on each axis, when all other axes have a score of 0 (the standard way), but this is different from what can be interpreted as the extremes of this space.

← Agreed. We changed the last part of the caption to:

“Drawings are shown for selected species (black bordered points) at the edges of the phylomorphospace spanned by the first three PCs.” (L. 578 f.)